# Investigation of the Effect of pH on the Adsorption–Desorption of Doxycycline in Feed for Small Ruminants

**DOI:** 10.3390/antibiotics12020268

**Published:** 2023-01-28

**Authors:** Rositsa Mileva, Tsvetelina Petkova, Zvezdelina Yaneva, Aneliya Milanova

**Affiliations:** Department of Pharmacology, Animal Physiology, Biochemistry and Chemistry, Faculty of Veterinary Medicine, Trakia University, 6000 Stara Zagora, Bulgaria

**Keywords:** adsorption, desorption, doxycycline, pH dependence, small ruminant feed

## Abstract

Orally administered tetracycline antibiotics interact with feed, which may impact their bioavailability and efficacy. Therefore, the pH-dependent adsorption of doxycycline and its interaction with feed for ruminants was studied in vitro. Adsorption experiments on animal feed (135 and 270 mg) with initial doxycycline concentrations of 35, 75, and 150 µg/mL were performed. Desorption experiments were conducted by agitation of a predetermined mass of doxycycline-loaded animal feed in PBS, at pH = 3.0, 6.0, and 7.4, to simulate changes in the gastrointestinal tract. Antibiotic concentrations were determined by LC-MS/MS analysis. The adsorption/desorption of doxycycline was described by mathematical models. Chemisorption with strong intermolecular interactions between the active functional groups of doxycycline and the organic biomass was found. The experimental release curve comprised three sections: initial prolonged 27–30% release (pH = 6.0), followed by moderate 56–59% release (pH = 3.0), and final 63–74% release (pH = 7.4). The sigmoidal model showed a considerable role of diffusion with an initial prevalence of desorption and a decreased desorption rate thereafter. The Weibull equation revealed an initial release stage followed by a lag time section and sustained release. The study of doxycycline adsorption by the animal feed proved a maximum 80% encapsulation efficiency and revealed initial diffusion followed by chemisorption. The highest release efficiency of 74% suggests high bioavailability of doxycycline after oral administration in ruminants.

## 1. Introduction

In modern animal husbandry, large numbers of animals are concentrated on a small area, which is a prerequisite for the occurrence of bacterial infections. Although many actions have been taken to improve animal welfare and replace antibiotics, infectious diseases occur which require adequate therapy with antibacterial drugs [1]. The problem of the selection and spread of bacterial resistance against antibiotics in industrial animal husbandry requires knowledge of the factors affecting the antibiotics’ bioavailability, which can decrease their efficacy. Oral administration of antibiotics with drinking water is the most often recommended route because the animals drink water even when they refuse to consume feed [2]. While subcutaneous or intramuscular administration of antibiotics is in most cases associated with predictable concentrations in the body, the oral route leads to complex interactions of the antibiotics with the microbiome and the contents of the gastrointestinal tract [3,4,5]. Therefore, precise use of antibiotics requires knowledge of a number of factors that may affect their absorption and the achievement of effective concentrations at the site of action, which is essential for the treatment of systemic infections [3]. 

Tetracyclines continue to be a group of antibiotics with significant use in veterinary practice, and new EU regulations define them as drugs with limited negative impact on resistance development and spread in the terms of co-selection [6]. The cited document emphasizes the requirements for their responsible use. They are most often applied orally, preferably with drinking water, in the mass treatment of farm animals [7]. Doxycycline is one of the preferable tetracyclines in the treatment of bacterial infections in animals. Its salts dissolve in water, and their solutions are stable for longer periods at acidic pH, while in alkaline pH they precipitate [8]. Chelation of tetracyclines with metal ions such as Ca^2+^, Al^3+^, and others reduces their bioavailability and efficacy. Compared to the older members of the group, doxycycline has a lower affinity for metal ions [9]. However, its ability to form chelated complexes with them has been observed [10]. This interaction is a prerequisite for the excretion of significant concentrations of tetracyclines in the environment through fecal masses and for their long retention in water and soil. The interaction of tetracyclines with feed masses and their adsorption on them in the gastrointestinal tract is also very important. It hinders the absorption and effectiveness of these antibiotics [3,4]. The non-specific binding rates of doxycycline to poultry feed were found to be 87.9 to 88.8%, respectively, at pH 2.5 and 6.5 [11]. In the cited study, it was found that the mixing of mycotoxin binders with the feed at effective doses did not affect the adsorption of doxycycline. An absence of interaction of doxycycline with mycotoxin binders and lack of effect on the bioavailability of the antibiotic were observed in pigs in another experimental setting [12]. These data suggest that the presence of nutrients in the gastrointestinal tract can significantly reduce the bioavailability of doxycycline in a wide pH range from 2.5 to 6.5. The presence of large amounts of nutrients in the rumen of ruminants could have a significant effect on the bioavailability of doxycycline. Additionally, the slow passage of food through the gastrointestinal tract of ruminants can affect their absorption. Taking into account the literature data for other animal species, the importance of doxycycline’s interaction with feed can be defined as significant, but the available literature about the interaction of doxycycline with feed for ruminants, depending on pH, is scarce.

Therefore, the aim of the present study was to investigate the adsorption of doxycycline, as a representative of the tetracyclines, on feed for ruminants and to characterize the interaction by simulating release conditions in different parts of the gastrointestinal tract via changing the pH. Mathematical models were used to describe the adsorption and desorption behaviors of doxycycline. 

## 2. Results and Discussion

In the current investigations, the effect of initial doxycycline concentration and animal feed mass on the adsorption efficiency of the solid phase was investigated (Figure 1a,b). The kinetics experimental data revealed a direct relationship between the antibiotic initial concentration and adsorption capacity, which is due to the greater number of organic molecules saturating the active sites of the biomaterial (Figure 1a). An increased mass of feed, however, was associated with lower capacity as a result of the smaller number of molecules occupying a greater number of adsorption sites (Figure 1b).

The goals of mathematical modelling of sorption/desorption processes are to provide an opportunity for prediction of the sorption behavior of a given system at different conditions or varying system parameters, to define the rate-limiting stage, and to reveal the mechanism of the adsorption/desorption process. 

### 2.1. Adsorption of Doxycycline

The kinetics experimental data of doxycycline adsorption on the animal feed were described by the pseudo-second-order kinetics model and the diffusion–chemisorption model. The values of the characteristic model parameters and correlation coefficients determined by linear regression analyses are presented in Table 1. 

The high values of the correlation coefficients (R^2^ > 0.9403) and the significantly close values of the experimental and model adsorption capacities q_e2_ and q_e_^DC^ for all studied doxycycline/animal feed experimental series determined the applicability of both models to describe the kinetics of the antibiotic adsorption by the animal feed. The series with doxycycline concentration C_0_ = 35 µg/mL and fodder mass w = 270 mg was characterized by the highest initial rate of adsorption (h = 510.91 µg/(mg.min)), which could be explained by the absence of competition between sorbate molecules due to their low number combined with the greater number of vacant active sites on/within fodder particles. Obviously, the increased initial doxycycline concentration and the reduced fodder mass were prerequisites for the lower initial adsorption rate of the experimental series with C_0_ = 75 µg/mL and 150 µg/mL and with w = 270 mg fodder mass. 

Considering the theoretical assumptions of the pseudo-second-order kinetics model that chemisorption is the operative reaction mechanism, the latter observations undoubtedly outline the significant role of chemical processes related to the formation of strong intermolecular interactions between the active functional groups of doxycycline and those of the organic biomass, as the rate-limiting stage of the process. 

Fodder, as a biosorbent, is characterized by inherent complex physical, chemical, and biological characteristics, making it necessary to test multiple kinetic models to achieve the best possible simulation. Such a multiple model is the diffusion–chemisorption model in which the rate of solid-phase concentration change (q_t_, µg/mg) is a function of the rate of mass transfer of the organic molecules from the fluid phase to the biosorption site, characterized by the rate constant (K_DC_, µg/(mg∙t^0.5^)), the equilibrium sorption capacity (q_e_, µg/mg), and the square root of time. The comparative analyses of the calculated rate constants (Table 1) reveal an increase in the diffusion–chemisorption rate with increasing doxycycline initial concentration up to 150 µg/mL. This trend is expected due to the increased concentration gradient developed between the inner and outer regions of the sorbent particles, which is the driving force for diffusion. Thus, the significant applicability of the diffusion–chemisorption model indicates that the remarkable role of film/intraparticle diffusion of the antibiotic molecules through the boundary layer surrounding the fodder particulates or within the internal pores, especially at the initial stages of the adsorption process, cannot be neglected. Based on these data, it can be concluded that the adsorption kinetics of doxycycline on fodder are expected to depend mainly on diffusion-limited processes, as affected by the heterogeneous distributions of active sites, functional groups, and pore sizes, and continual partitioning of antibiotic molecules between the dissolved state and fixed state of adsorption [13].

### 2.2. Desorption of Doxycycline

The in vitro desorption experiments aimed to simulate the behavior of doxycycline-loaded feed in the gastrointestinal tract of ruminants, which, in turn, could allow the assessment of the negative health effects arising from the consumption of antibiotic-containing fodder. The desorption processes were modelled in an attempt to explicate the rate-limiting steps and to select an appropriate release model which can quantify the effect of changing solution and sorbent parameters and can aid in the eventual development of predictive models that would enable process design and in vitro behavior analyses with minimal experimentation. As the accuracy of fit is paramount in the development of predictive models, the error functions SSE, MSE, and RMSE and correlation coefficients were also determined.

The experimental doxycycline in vitro release from doxycycline-loaded fodder was studied in a simulated physiological medium to assess the effect of pH on the extent of antibiotic desorption in the gastrointestinal tract of ruminants. Two series of experiments were conducted. The first series was run with the mass of doxycycline-loaded fodder w = 133.3 mg containing 11.03 µg/mg of the antibiotic in the solid phase (Figure 2a). The second series was performed with the mass of doxycycline-loaded fodder w = 137.8 mg containing 41.6 µg/mg of the antibiotic in the solid phase (Figure 2b). The solid-phase concentrations of doxycycline were calculated on the basis of the quantity of antibiotic encapsulated by the feed. All the data were derived from the experiments performed in the current study. The extent of the in vitro antibiotic desorption at pH = 6.0 (simulating the pH in the rumen), at pH = 3.0 (simulating the pH in the abomasium), and at pH = 7.4 (simulating the pH in the small intestine) was investigated.

The experimental kinetics release curves, characterized by a pulsatile release mode, comprised three well-defined major sections (Figure 2a,b). The initial part resembled the conditions in the rumen and showed prolonged 27–30% release at pH = 6.0 for 1.5 h of incubation. Moderate release, up to 56–59%, was observed at pH = 3.0, which simulated the pH in the abomasium. The last part of the curve presented the final 63–74% release at pH = 7.4, close to the pH in the small intestines. The suggested concept for the release mechanism in the last stage is the infiltration of tiny antibiotic-loaded feed particles into the mucus, resulting in gradual disintegration, facilitated doxycycline release, and subsequent permeation through the paracellular pathway to the bloodstream.

Four single-, two-, and three-parameter release kinetics mathematical models—Higuchi, Korsmeyer–Peppas, Weibull, and the sigmoidal model—were applied to describe the experimental desorption results. The values of the calculated model parameters and error functions obtained by linear and nonlinear regression analyses are presented in Table 2.

The Higuchi model assumes that the drug release occurs predominantly through Fickian diffusion and has typically been observed in drug-carrier systems with constituents of a hydrophobic nature. The low value of R2 and the high values of the other error functions in the present study indicated the limited applicability of this model for the current experimental data.

The Korsmeyer–Peppas model describes simultaneously the following release mechanisms: water diffusion into the biomatrix, matrix swelling, and dissolution/relaxation of the matrix [14]. The release exponent n of the Korsmeyer–Peppas model describes the drug release mechanism: Fick diffusion for n = 0.5 and non-Fickian diffusion for 0.5 < n < 1. The values of the parameter in this study (n = 0.815–0.982) indicate anomalous or non-Fickian antibiotic release including diffusion and relaxation effects; thus, an appropriate model such as the sigmoidal model, which could describe such complex behavior, was required. The release (transport) constant (a) provides information on the drug formulation, such as the structural characteristics of the carrier. The identical characteristics of the fodder explain the commensurable values of the parameter a for both studied series.

In the Weibull equation, the parameter T is a location parameter denoting the lag time before the onset of the drug release procedure, while b describes the shape of the dissolution curve progression. For b = 1, the shape of the curve corresponds to an exponential profile, which coincides with the mode of the experimental release curve. Obviously, the modes of the experimental release curves generally include an initial release stage followed by a lag-time section and then sustained release. Lag time in pharmacokinetics is associated with the finite time necessary for a drug to enter the central circulation after extravascular administration. This parameter describes the absorption phase and depends on the drug dissolution/release processes from the delivery system and on drug migration to the surface of the sorbent [15,16].

Due to the two clearly defined separate regions (region 1: until 1.5 h and region 2: from 1.5 h to 3 h) on both experimental kinetics release curves, a modified modelling methodology was undertaken. The sigmoidal model was applied separately to each of the regions due to the deviations in the release behavior of the antibiotic in the rumen as compared to that in the abomasum and the small intestine, which provoked different modes of the respective curves. The comparative analyses of the experimental and model data and the values of error functions proved the better applicability of the sigmoidal model, as it had the highest average R2 and the lowest average SSE, MSE, and RMSE values for both experimental series. Moreover, the model release curves practically coincided with the experimental ones for both studied series (Figure 3a,b).

The values of the diffusional constant k_s1_, the relaxation constant k_s2_, and the diffusional exponent n_s_ enable the assessment of the contributions of relaxation and diffusion mechanisms within the different stages of the release process. The data in Table 2 outline that the values of these constants for each of the regions are commensurable, which, in turn, indicates the relative contribution of both mechanisms within each region. However, the significantly higher values of the kinetic constants k_s1_ for region 1 as compared to k_s2_ for region 2 are indicative of the considerable role of diffusion in simulated rumen medium. Obviously, the role of relaxation as a limiting mechanism of doxycycline release from fodder is also significant. This could be explained by the chemical composition of fodder, namely, the high content of a variety of biopolymers (starch, hemicellulose, cellulose, lignins, etc.) that are susceptible to polymer chain relaxation caused by mechanical stress (agitation) and/or pH changes. The inflexion points registered at approximately 28% and 31% equilibrium release outline an initial stage where the rate of desorption exceeds that of sorption and a second region characterized by a decreased desorption rate. The probable formation of compact supramolecular networks comprising doxycycline molecules and water dipoles via intermolecular hydrogen bonding could also evoke sigmoidal behavior during the in vitro release process [17]. 

In conclusion, the adsorption study of doxycycline by animal feed proved a maximum 80% encapsulation efficiency of the fodder towards the antibiotic. Similar results were reported for interactions between doxycycline and feed for broiler chickens and pigs [11,12]. High non-specific binding of doxycycline (around 88%) to the components of the feed matrix was observed, which could have impact on its bioavailability [11]. The capacity for its adsorption on the feed was studied in in vitro experiments, conducted in buffered solutions with or without feed. These tests were performed at a specific acidic or alkaline pH and more often at neutral pH [18,19]. Our results, for the first time, describe interactions between doxycycline and feed for small ruminants, and they are in accordance with the observed dependence of the processes of adsorption and desorption of polar substances on the pH of the medium, which varies in different parts of the gastrointestinal tract [20,21]. Orally administered drugs pass through different parts of the digestive tract, where the values of pH vary from acidic in the stomach (3.5–5–6.5, depending on the species) to alkaline in the gut (7.5–7.8). Data from the modeling of doxycycline adsorption/desorption demonstrated the need to study these processes at different pH in order to adequately assess the presence of free drug molecules available for absorption. The limitations of the current investigation concern the use of feed specific for lactating sheep, while some changes in the proportions and the type of feeding of the animals can lead to variations in doxycycline–feed interaction. Although the metabolism of doxycycline is negligible, it would be of interest to discover how the addition of specific enzymes to the simulated medium affects the adsorption/desorption of the antibiotic in the feed for small ruminants.

## 3. Materials and Methods

### 3.1. Drugs and Reagents

Doxycycline hyclate with purity ≥98% (HPLC grade, Sigma-Aldrich, St. Louis, MO, USA) and crystalline oxytetracycline hydrochloride ≥95% were used as analytical standards. HydroDoxx 500 mg/g Oral Powder (Huvepharma, Sofia, Bulgaria) was used to prepare solutions for adsorption experiments. The following reagents were used for extraction and for the further analysis of the drugs: trifluoroacetic acid (99.5%) (Fisher Chemical, Waltham, MA, USA), LC/MS grade acetonitrile OPTIMA^®^ (Fisher Chemical), LC-MS grade methanol (CHROMASOLV LC-MS, Honeywell, Seelze, Germany), ~98% formic acid for mass spectrometry (Honeywell Fluka™, Seelze, Germany), and water for chromatography (LC-MS Grade, LiChrosolv^®^, Merck KGaA, Darmstadt, Germany). The reagents PBS (pH = 7.4) and HCl (ACS reagent, 37%) used for the in vitro release experiments were supplied by Sigma.

### 3.2. Experimental Procedure and Adsorption/Desorption Study

The feed used in the current experiments consisted of 80% hay (72% meadow hay and 8% alfalfa hay) and 20% concentrated feed for lactating domestic sheep (*Ovis aries*) with 18% crude protein (HL-TopMix, Sliven, Bulgaria). The concentrated feed contained maize, wheat, soybean meal, sunflower meal, maize fodder, protected fat, vitamin premix, and micro- and macro-elements, without copper. The analytical content was 18% crude protein, 7.5% crude fiber, and 3.2% crude fat. The amounts of coarse fodder and concentrated feed were selected according to the practices for feeding lactating sheep. The feed was milled and well-mixed before the experiments. The amount of doxycycline hyclate used to prepare the experimental solutions was calculated on the basis of a sheep with a body weight of 55 kg and an oral dose of 20 mg/kg doxycycline, according to the data from our previous study [22]. The concentrations of the antibiotic solutions were calculated by taking into account the total dose of the antibiotic and the volume of the contents of the rumen in small ruminants [23,24]. Based on these calculations, doxycycline solutions were prepared in Milli-Q (Evoqua Water Technologies, Pittsburgh, PA, USA) water at 35, 75, and 150 µg/mL, corresponding to low, medium, and high concentrations, respectively. The values of pH at different segments of the gastrointestinal tract were taken into account, and experiments were performed at pH 3.0, 6.0, and 7.4 [23]. 

Five series of adsorption experiments with initial doxycycline concentrations C_0_ = 35, 75, and 150 µg/mL, mass of animal feed (coarse/concentrated fodder = 3/2 *w/w*) w = 135 and 270 mg, and solution volume V = 50 mL at pH = 6 and T = 37 °C were conducted in batch mode for 3 h. The adsorption capacity of the feed (qt, µg/mg) was determined by the formula
(1)$$q_{t}=\frac{\left(\mathrm{Co}-\mathrm{Ct}\right)\cdot w}{V}$$
where Ct is the doxycycline concentration in the liquid phase at time t, µg/mL.

Desorption experiments were conducted by agitation of a predetermined mass of doxycycline-loaded animal feed in simulated gastrointestinal medium without enzymes, comprising PBS pH = 7.4 adjusted to pH = 3.0 and 6.0 with 1 M HCl, for 3 h at T = 37 °C in a Digital Waterbath WNB 22 (Memmert GmbH, Büchenbach, Germany). Samples were taken at predetermined time intervals, and fresh medium of an equal volume was added to restore the total amount of the medium. The release efficiency (E, %) was calculated by
(2)$$E=\frac{\mathrm{amount}\;\mathrm{of}\;\mathrm{released}\;\mathrm{substance}}{\mathrm{total}\;\mathrm{amount}\;\mathrm{of}\;\mathrm{encapsulated}\;\mathrm{substance}}\times 100\%$$

Blanks containing no antibiotic and replicates of each adsorption/desorption point were used for each series of experiments. The concentration of doxycycline in the liquid phase was determined by LC-MS/MS (Agilent 6400c, Agilent Technologies, Santa Clara, CA, USA). Standard solutions were prepared in the same matrix as blank samples. Blank matrix was used for dilution of the samples.

### 3.3. LC-MS/MS Analysis of Doxycycline in Samples Containing Animal Feed

The extraction of doxycycline from samples containing 135 or 270 mg animal feed (coarse/concentrated fodder = 3/2 *w/w*) in 50 mL simulated gastrointestinal fluid was carried out by the method previously described [25]. Dilutions of the samples were prepared in the same matrix which did not contain the antibiotic. Several serial dilutions 1:10 *v:v* were prepared from every sample so that the suitable dilution was used for the determination of the concentrations. The extraction was performed as follows: 500 μL samples were added to 10 μL oxytetracycline as an internal standard (at final concentration 0.1 μg/mL) and 65 μL trifluoroacetic acid (TFA). The samples were vortexed for 1 min and centrifuged at 10,800× *g* for 10 min at 22 °C. After this step, the supernatant was filtered through 0.22 μm syringe filters and transferred into injection vials. Then, 5 μL of the filtrate was injected into the LC-MS/MS system. The calibration curve was prepared in the same matrix. The final concentrations of doxycycline were 0, 0.01, 0.05, 0.1, 0.25, 0.5, 0.75, and 1 μg/mL. 

The LC-MS/MS method was used to determine the concentration of doxycycline. An Agilent 6460c triple-quadrupole mass spectrometer with AJS technology was used for the analysis. The liquid chromatography (LC) system consisted of a 1260 Infinity II quaternary pump and a 1260 Infinity II Vial Sampler. A Poroshell 120 EC C18 column (4.6 mm i.d. 100 mm, 2.7 m, Agilent Technologies, Santa Clara, CA, USA) was used at a temperature of 40 °C for chromatographic separation of the tetracyclines, using doxycycline as the target compound and oxytetracycline as an internal standard. The mobile phases for LC-MS analysis were a) 0.1% formic acid in ultrapure water and b) 100% acetonitrile. Gradient elution was used for separation of the compounds at a flow rate of 0.3 mL.min-1. The gradient was as follows: 0–0.5 min (90% A, 10% B), 0.50–8 min (from 90% A, 10% B to 2% A, 98% B), and 8–12 min (2% A, 98% B). The injection volume was 5µL. The total run time was 12 min with a post-run of 5 min. The gas temperature was 350 °C, the drying gas (nitrogen) flow rate was 12 L/min, the nebulizer gas (nitrogen) pressure was 45 psi, the sheath gas (nitrogen) temperature was 400 °C, and the sheath flow rate was 12 L/min. The capillary voltage, nozzle voltage, and dwell time were 4000 V, 500 V, and 200 ms, respectively. The qualifying ion for doxycycline was 445.1 m/z, and that for oxytetracycline was 461.1 m/z. The quantifying ions for these substances were 410/428.1 m/z and 443.1/444 m/z, respectively. The standard curve was linear (R^2^ = 0.9992) between 0.01 and 1 μg/mL. The values of mean accuracy and inter-day and intra-day precision were 95.30 ± 9.17%, 1.16–11.23, and 0.58–2.32%, respectively. The values of LOD and LOQ were 1.12 ng/mL and 3.41 ng/mL.

### 3.4. Mathematical Modelling

The experimental adsorption data were described by the pseudo-second-order kinetics and diffusion–chemisorption models [26]. The desorption behavior of the studied system was modelled by four release kinetics models: Higuchi, Korsmeyer–Peppas, Weibull, and the sigmoidal model [27]. Linear and nonlinear regression analyses were applied.

### 3.5. Statistical Analysis

The statistical analyses of the experimental results, the values of the correlation coefficients R2, and the error functions (sum of the squared errors (SSE), mean squared error (MSE), and root mean square error (RMSE)) were determined by linear/nonlinear regression analyses using XLSTAT statistical software for Excel (Microsoft Corporation, Washington, USA). 

## 4. Conclusions

The data in the current study, for the first time, describe the interactions between doxycycline and feed for small ruminants, with the aim to obtain more information about possible effects on the bioavailability of the antibiotic after its oral administration. The results from our proposed mechanism of adsorption based on the applied mathematical models comprise initial diffusion followed by chemisorption. The in vitro release behavior of doxycycline followed a sigmoidal mode comprising three sections: initial prolonged release in the rumen, moderate release in the abomasium, and final release in the small intestine. The highest release efficiency of 74% in simulated physiological medium indicates high bioavailability of the drug after oral administration in small ruminants. Initial adsorption of the antibiotic on the fodder was followed by almost complete desorption at pH 7.4, which could contribute to significant absorption of doxycycline and the achievement of therapeutic levels in central circulation.

## Figures and Tables

**Figure 1 antibiotics-12-00268-f001:**
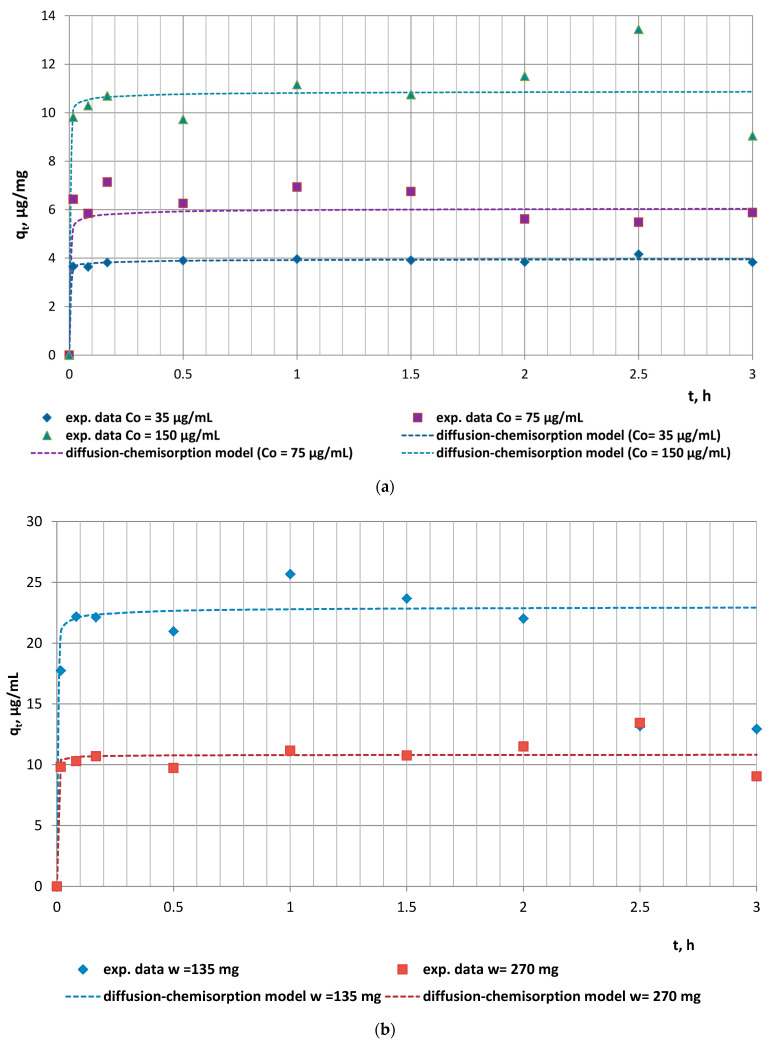
Experimental encapsulation kinetics results vs. diffusion–chemisorption model data: (**a**) effect of initial concentration (w = 270 mg); (**b**) effect of animal feed mass (C_0_ = 150 µg/mL).

**Figure 2 antibiotics-12-00268-f002:**
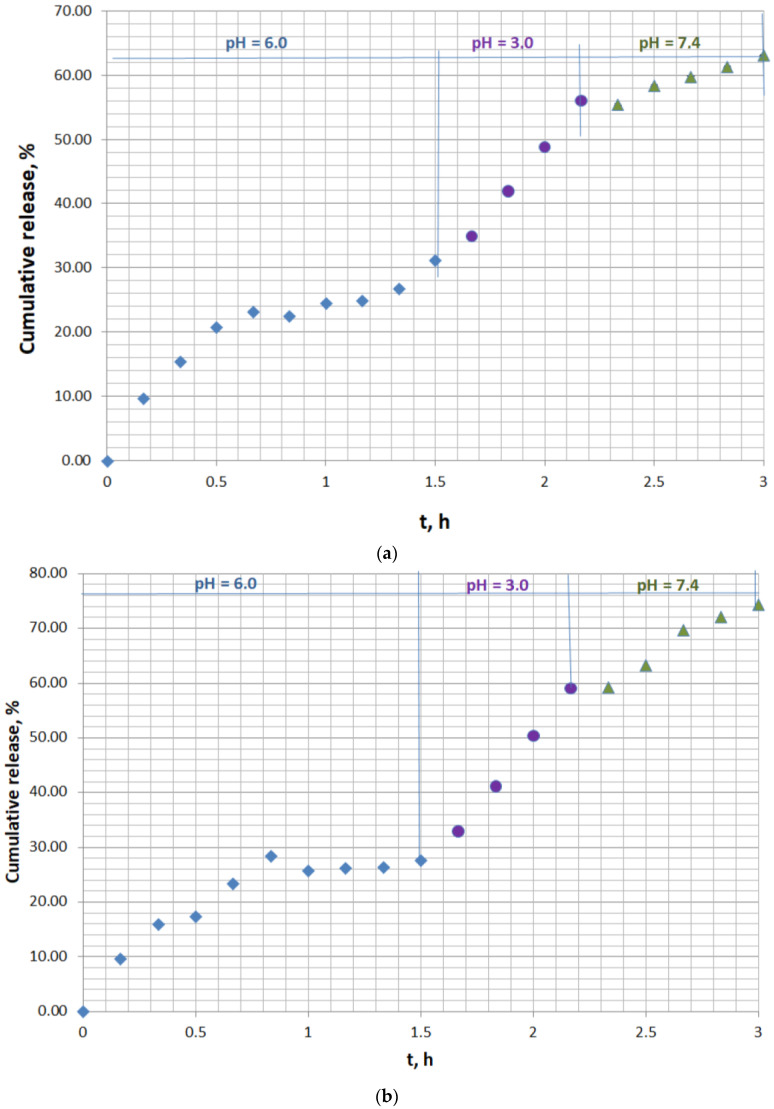
In vitro kinetics release curve of doxycycline from animal feed in simulated gastrointestinal and stomach medium: (**a**) solid doxycycline concentration 11.03 µg/mg, doxycycline-loaded fodder w = 133.3 mg; (**b**) solid doxycycline concentration 41.6 µg/mg, doxycycline-loaded fodder w = 137.8 mg.

**Figure 3 antibiotics-12-00268-f003:**
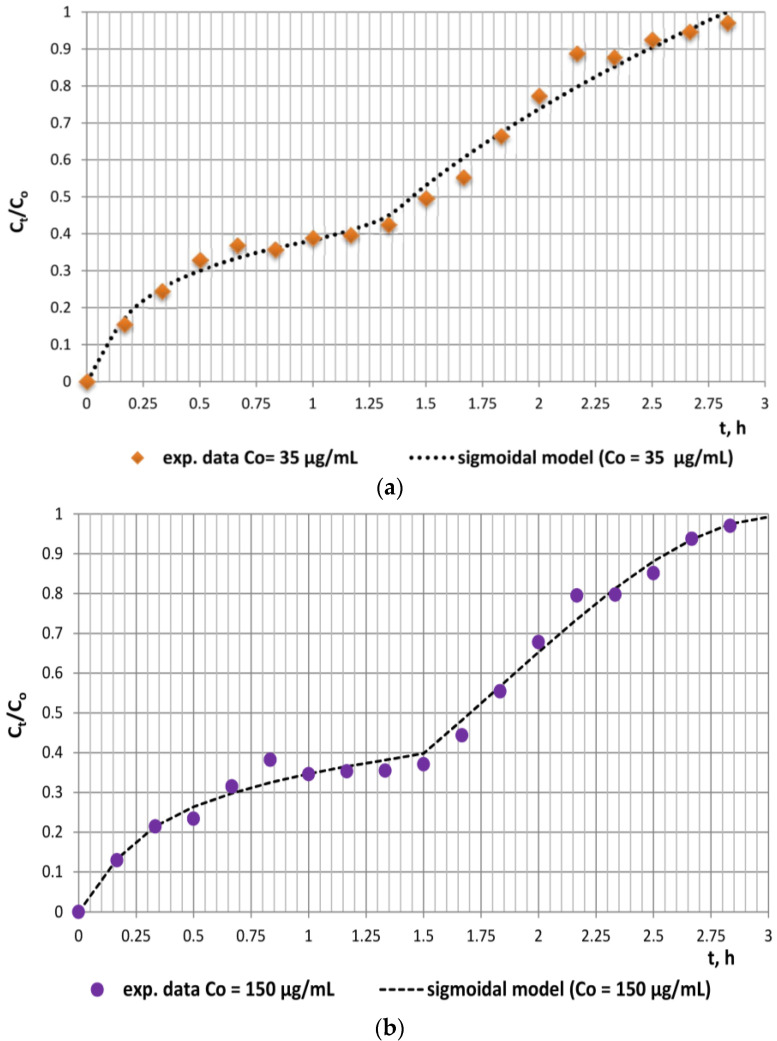
Applicability of the sigmoidal model to the experimental in vitro release data: (**a**) solid doxycycline concentration 11.03 µg/mg, doxycycline-loaded fodder w = 133.3 mg; (**b**) solid doxycycline concentration 41.6 µg/mg, doxycycline-loaded fodder w = 137.8 mg.

**Table 1 antibiotics-12-00268-t001:** Values of the model parameters and error functions of the applied adsorption kinetics models to the system doxycycline/animal feed.

Mathematical Model	Model Parameters			
	Effect of Initial Concentration (w = 270 mg)			
	C_o_ = 35 µg/mL	C_o_ = 75 µg/mL		C_o_ = 150 µg/mL
Pseudo-second-order kinetics model $\frac{1}{q_{t}}=\frac{1}{k_{2}\times q_{e}^{2}}+\frac{1}{q_{e}}\times t$	q_e2_ = 3.934 µg/mg k_2_ = 129.87 h = 510.91 µg/(mg∙min) R^2^ = 0.9972	q_e2_ = 5.685 µg/mg k_2_ = 3.291 h = 18.71 µg/(mg∙min) R^2^ = 0.9915		q_e2_ = 10.515 µg/mg k_2_ = 3.015 h = 31.70 µg/(mg∙min) R^2^ = 0.9464
Diffusion–chemisorption model $\frac{t^{0.5}}{q_{t}}=\frac{1}{q_{e}}\times t^{0.5}+\frac{1}{K_{DC}}$	K_DC_ = 357.14 q_e_^DC^ = 3.962 µg/mg R^2^ = 0.9968	K_DC_ = 277.78 q_e_^DC^ = 6.11 µg/mg R^2^ = 0.9781		K_DC_ = 2000.00 q_e_^DC^ = 10.846 µg/mg R^2^ = 0.9403
	**Effect of animal feed mass (C_0_ = 150 µg/mL)**			
	**w = 135 mg**		**w = 270 mg**	
Pseudo-second-order kinetics model	q_e2_ = 15.748 µg/mg k_2_ = 0.4337 h = 6.83 µg/(mg∙min) R^2^ = 0.8853		q_e2_ = 10.515 µg/mg k_2_ = 3.015 h = 31.70 µg/(mg∙min) R^2^ = 0.9464	
Diffusion–chemisorption model	K_DC_ = 1666.66 q_e_^DC^ = 23.095 µg/mg R^2^ = 0.9916		K_DC_ = 2000.00 q_e_^DC^ = 10.846 µg/mg R^2^ = 0.9403	

k_2_: pseudo-second-order rate constant; h: initial rate of adsorption, µg/(mg.min); KDC: rate constant in the diffusion–chemisorption model.

**Table 2 antibiotics-12-00268-t002:** Values of the model parameters and error functions of the applied in vitro release kinetics models to the doxycycline/animal feed system in simulated gastrointestinal and stomach medium.

Mathematical Model	Model Parameters		Error Functions		Regression Analyses
	Series 1	Series 2	Series 1	Series 2	
Higuchi $C_{t}=k_{H}\times t^{0.5}$	k_H_ = 4.33	k_H_ = 18.56	R^2^ = 0.9139	R^2^ = 0.8672	Linear regression
Korsmeyer–Peppas $\frac{C_{t}}{C_{o}}=a\times t^{n}$	a = 0.083 n = 0.815	a = 0.079 n = 0.982	R^2^ = 0.9139 SSE = 0.003 MSE = 1.6 × 10^−4^ RMSE = 0.013	R^2^ = 0.9139 SSE = 0.004 MSE = 2.3 × 10^−4^ RMSE = 0.015	Nonlinear regression
Weibull $C_{t}=C_{o}\times \left[1-e^{-\frac{{\left(t-T\right)}^{b}}{a}}\right]$	at b = 1 C_o_ = 97.19 a_w_ = 0.022 T = 0.291	at b = 1 C_o_ = 21.6 a_w_ = 0.26 T = 0.469	R^2^ = 0.962 SSE = 3.101 MSE = 0.194 RMSE = 0.440	R^2^ = 0.960 SSE = 63.085 MSE = 3.943 RMSE = 1.986	Nonlinear regression
Sigmoidal $\frac{C_{t}}{C_{e}}=k_{s1}\times {\left(t-l\right)}^{n_{s}}+k_{s2}\times {\left(t-l\right)}^{n_{s}}$	Region 1 k_s1_ = 27.063 n_s_ = 0.004 k_s2_ = 27.446	Region 1 k_s1_ = 130.31 n_s_ = 0.001 k_s2_ = 130.657	Region 1 R^2^ = 0.958 SSE = 0.002 MSE = 0.001 RMSE = 0.025	Region 1 R^2^ = 0.904 SSE = 0.005 MSE = 0.001 RMSE = 0.033	Nonlinear regression
	Region 2 k_s1_ = 11.011 n_s_ = 0.057 k_s2_ = 11.267	Region 2 k_s1_ = 0.190 n_s_ = 2.111 k_s2_ = 0.009	Region 2 R^2^ = 0.964 SSE = 0.020 MSE = 0.002 RMSE = 0.047	Region 2 R^2^ = 0.984 SSE = 0.009 MSE = 0.001 RMSE = 0.033	

k_H_: Higuchi release rate constant; a: release constant in the Korsmeyer–Peppas model; aw: time process parameter in the Weibull model; T: lag time; b: shape parameter in the Weibull model, characterizing the curve as exponential at b = 1; k_s1_, k_s2_: sigmoidal model release constants; n_s1_, n_s2_: sigmoidal model exponents.

## Data Availability

Not applicable.

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
