# Peer review of "Investigation of the Effect of pH on the Adsorption–Desorption of Doxycycline in Feed for Small Ruminants"

_antibiotics, 2023, doi:10.3390/antibiotics12020268_

Round 1

Reviewer 1 Report

In this paper, pH dependent adsorption of doxycycline and its interaction with the feed for ruminants was investigated in vitro. The mathematical models were used to describe the adsorption and desorption behaviour of doxycycline.This paper is informative and complete. But the following problems need to be solved before acceptance.

1. The figures in this manuscript should be more clearly with sufficient resolution.

2. The caption in figure. 1A should be corrected. “diffussion” should be changed to “diffusion”.

3. In the “2. Results and discussion”, it would be more clearly if the secondary headings were added.

4. Some formal errors should be corrected. Such as “ in vitro” should be in italic.

Author Response

Thank you for the suggestions, we tried to fulfill all the criteria according to the remarks.

Kind regards,

The authors

Comments and Suggestions for Authors

Reviewer 1

In this paper, pH dependent adsorption of doxycycline and its interaction with the feed for ruminants was investigated in vitro. The mathematical models were used to describe the adsorption and desorption behaviour of doxycycline. This paper is informative and complete. But the following problems need to be solved before acceptance.

  1. The figures in this manuscript should be more clearly with sufficient resolution.

Answer: The pictures were prepared in tiff format with dimensions that exceeded the requirements: 1200 dpi and with width >8000 pixels and height > 5000 pixels. They were sent as original files and we hope that in the final version they will look better because the final proof will be prepared in professional way.

 The caption in figure. 1A should be corrected. “diffussion” should be changed to “diffusion”.

Answer: Thank you for this remark, the revision has been performed.

  1. In the “2. Results and discussion”, it would be more clearly if the secondary headings were added.

Answer: Subheadings were used at line 89: “2.1. Adsorption of doxycycline” and at line 137: “2.2. Desorption of doxycycline

  1. Some formal errors should be corrected. Such as “ in vitro” should be in italic.

Answer: The correction was done.

Reviewer 2 Report

The study is well conducted and tackles the need to solve a problem in the veterinary medicine practice 

1-    Title is vague, what do you mean by sorption? I suggest changing the title to “Investigating the effect of pH-dependent absorption on doxycycline with ruminant drug-food interaction for veterinary use”

2-    Abstract should be structured based on MDPI format: background, method, result, conclusion

3-    Please add the limitation of the study 

Author Response

Thank you for the remarks, we tried to fulfill the required changes and we checked again the instructions to authors. We hope that we present the revision in a proper form.

Kind regards,

The authors

Comments and Suggestions for Authors

Reviewer 2

The study is well conducted and tackles the need to solve a problem in the veterinary medicine practice

 1- Title is vague, what do you mean by sorption? I suggest changing the title to “Investigating the effect of pH-dependent absorption on doxycycline with ruminant drug-food interaction for veterinary use”

Answer: We tried to clarify the title and we hope that the following title will be accepted by the Reviewer: “Investigation of the effect of pH on the adsorption-desorption of doxycycline in feed for small ruminants”

2- Abstract should be structured based on MDPI format: background, method, result, conclusion

Answer: During the preparation of the manuscript we followed the instruction for the authors “The abstract should be a single paragraph and should follow the style of structured abstracts, but without headings”. We also checked the arrangement of the abstract in the published articles. Our abstract contains information related to the background and aim of the study, methods, results and conclusion. Therefore, we hope that it will be accepted to keep the abstract as it is.

3- Please add the limitation of the study 

Answer: Thank you for this remark. The following information about the limitation of the study was included at lines 257-263: “The limitations of the current investigation concern the use of feed specific for lactating sheep, while some changes in the proportions and the type of feeding of the animals can lead to variations in doxycycline–feed interaction. Although the metabolism of doxycycline is negligible, it would be of interest to discover how the addition of specific enzymes to the simulated medium affects the adsorption/desorption of the antibiotic in the feed for small ruminants.”

Reviewer 3 Report

The authors should consider the followings: 

The authors may provide the results of the validated stability studies (4c, physio, pH, room temperature, if available) to the article. 

The authors should discuss the limitions of the study, in the discussion section. 

The authors should state clearly in the abstract and or conlcusion part, for the novel findings in the study. 

The authors may give details rationales of choosing the said conc of Doxycycline in Fig 3a and 3b. 

Please specify the target ruminants (family and or species) in the study, with defined physiological assumption. 

The authors may give details rationales of choosing the said conc of Doxycycline in Fig 2a and 2b.  Did the authors validate the dilution efficiency by the said blank solution?

Please show the validation results in supplementary section. 

Please clarify how many batches of animal feed were used in the study. Did the author observe significant difference of the results, if using different batches of the feed.

Author Response

Comments and Suggestions for Authors

Reviewer 3

The authors should consider the followings: 

The authors may provide the results of the validated stability studies (4c, physio, pH, room temperature, if available) to the article. 

Answer: The method for doxycycline determination has been validated for HPLC analysis and for LC-MS/MS analysis in our lab and the results were published elsewhere (DOI:https://doi.org/10.2478/macvetrev-2019-0016; https://doi.org/ 10.3390/pharmaceutics14112440). Therefore, additional stability studies were not performed and the analysis were started on the day after experiment for every batch and were run within 3-4 days. The temperature conditions at the working room were well controlled and were kept at 22°C. The conditions in the autosampler and the temperature of the column during the analysis were always the same and were described in the materials and methods.

 The authors should discuss the limitions of the study, in the discussion section. 

Answer: Thank you for this remark. The following information about the limitation of the study was included at lines 257-263: “The limitations of the current investigation concern the use of feed specific for lactating sheep, while some changes in the proportions and the type of feeding of the animals can lead to variations in doxycycline–feed interaction. Although the metabolism of doxycycline is negligible, it would be of interest to discover how the addition of specific enzymes to the simulated medium affects the adsorption/desorption of the antibiotic in the feed for small ruminants.”

The authors should state clearly in the abstract and or conlcusion part, for the novel findings in the study. 

Answer: We are thankful for this suggestion made by the Reviewer. Some changes in the manuscript were performed so that the reader could understand what was done for first time with this work (lines 249-250 and lines 363-365).

 The authors may give details rationales of choosing the said conc of Doxycycline in Fig 3a and 3b.

Answer:  The solid-phase concentrations of doxycycline were calculated on the basis of the quantity of antibiotic encapsulated by the feed. All the data were derived from the experiments performed in the current study. (Lines 155-158)

 Please specify the target ruminants (family and or species) in the study, with defined physiological assumption.

Answer: The required information has been included at line 279: “domestic sheep (Ovis aries)”.

The authors may give details rationales of choosing the said conc of Doxycycline in Fig 2a and 2b. Did the authors validate the dilution efficiency by the said blank solution?

Answer: The explanation for the used concentrations is the same as for Figure 3, as has been described above.

The standard solutions for preparation of the calibration curve were made in blank solution, obtained after incubation of feed (135 or 270 mg, according to the series of the experiments) in PBS for 3 h at T = 37°C. Due to very high concentrations of doxycycline it was impossible to analyze the concentration in undiluted samples. Every sample was run several times with different dilutions. We started with the highest dilution in order to have an idea about the most suitable dilution. The final dilution was chosen so that the concentrations to be within the range of the standard curve. The standard curves were prepared in the same way as the samples. The highest concentration (stock solution) was prepared in the same medium as the medium used for the samples.

Please show the validation results in supplementary section.

Answer: The validation results for the method were presented in the main text and they concern PBS with pH 7.4 and unfortunately we do not have additional validation results. The validation procedure showed that the method was stable and Intra-day and Inter-day precision for 50 ng/ml were 1.19% and 7.66%; and for 500 ng/ml were 0.19% and 3.03%, respectively. The accuracy for these concentrations were 91.82±7.38 and 96.59±3.14, respectively. We used the same blank matrix for dilution of the samples and we observed that there is no matrix effect when calibration curves were built with the same matrix as that used for dilution of the samples. Additionally, we followed as a guidance for our work the previously published articles which found that doxycycline was stable at pH<8, including at pH=2.5 and pH=6.5 (doi: 10.1021/acs.jafc.5b06084 and doi: 10.1080/19440049.2018.1501163). The applied extraction procedure in the cited articles and in the current investigation are very similar and we did not use extreme pH>8 (our settings was between pH = 7.4 to pH = 3).

 Please clarify how many batches of animal feed were used in the study. Did the author observe significant difference of the results, if using different batches of the feed.

Answer: We used well milled and mixed feed for sheep which contained hay and concentrated feed as described in material and methods. The concentrated feed was obtained from one of the biggest company in Bulgaria specialized for feed production which provided the description of the content of the batches which are standardized. The hay is prepared for the whole season and it was obtained from the University farm, where sheep are reared. Higher amount of hay and concentrated feed than necessary for the experiments were obtained and milled and they were very well mixed. The experiments were run with this batch of feed that was used during several months within one year in a sheep farm and there were no differences between repetition of the runs of experiments. Therefore, we pointed this as a limitation of the study.

Once again thank you for the critical remarks which helped for clarification and improvement of the presented investigation. We tried to revise the manuscript according to the remarks and we hope that the revision is at acceptable level. The manuscript has been checked by English editor (Please, see the attached file).
